

# Large contribution of organics to condensational growth and formation of cloud condensation nuclei (CCN) in remote marine boundary layer

Guangjie Zheng[1,2], Chongai Kuang[2], Janek Uin[2], Thomas Watson[2], and Jian Wang[1,2*]

[1] Center for Aerosol Science and Engineering, Department of Energy, Environmental and Chemical Engineering, Washington University in St. Louis, Missouri, USA
[2] Environmental and Climate Science Department, Brookhaven National Laboratory, Upton, New York, USA

*Correspondence to*: J.W. (jian@wustl.edu)

**Abstract.**

Marine low clouds strongly influence global climate, and their radiative effects are particularly susceptible to the concentration of cloud condensation nuclei (CCN). One major source of CCN is condensational growth of pre-CCN particles, and sulfate has long been considered the major condensing species in remote marine boundary layer. While some studies suggested that secondary organic species can contribute to the particle growth, its importance remains unclear. Here we present the first long-term observational evidence that organics play an important role in particle growth over remote oceans. To the contrary of traditional thinking, sulfate dominated condensational growth for only a small (~18%) fraction of the 62 observed growth events, even fewer than the organic-dominated events (24%). During most (58%) growth events, the major condensing species included both organics and sulfate. Potential precursors of the secondary organics are volatile organic compounds from ocean biological activities and those produced by the air-sea interfacial oxidation. Our results indicate that the condensation of secondary organics contributes strongly to the growth of pre-CCN particles, and thereby the CCN population over remote oceans.



## 1. Introduction

Marine low clouds play an important role in global climate system (Wood, 2012), and their properties and radiative effects are very sensitive to the concentration of cloud condensation nuclei (CCN) (Carslaw et al., 2013; Rosenfeld et al., 2019). Condensational growth of pre-CCN particles (i.e., particles that are too small to form cloud droplets) (Hoppel et al., 1990; Pierce and Adams, 2006) is one major source of CCN in remote marine boundary layer (MBL) (Pierce and Adams, 2006; Yu and Luo, 2009; Sanchez et al., 2018), and is likely the dominant one in late-spring to fall (Zheng et al., 2018). Over open ocean,

dimethyl sulfide (DMS) is the dominant biogenic volatile organic compound (VOC). The major oxidation products of DMS are sulfur dioxide ($SO_2$) and methanesulfonic acid (MSA) (Andreae et al., 1985). Further oxidation of $SO_2$ produces sulfuric acid ($H_2SO_4$), which readily condenses onto existing particles and participates in the formation of new particles (Kulmala et al., 2000). It has long been recognized that sulfate produced from DMS oxidation is a major species for particle condensational growth in the remote marine environment (Sanchez et al., 2018). Earlier studies (Willis et al., 2016; Kerminen and Wexler,

1997; Karl et al., 2011) suggest that MSA may also contribute to the growth of pre-CCN particles and thus the formation of CCN. However, the effect of MSA condensation on marine CCN concentration remains unclear. Model simulated effects range from negligible (e.g., a few percent) to significant (~20%) depending on the assumption of MSA volatility and the geographic location (Hodshire et al., 2019).

It has been suggested that in the remote MBL, secondary organics produced from two types of non-DMS VOCs can contribute substantially to particle condensational growth. The first type of VOCs, including isoprene, monoterpenes, and aliphatic amines (Facchini et al., 2008; Dall'Osto et al., 2012; Willis et al., 2017), is related to ocean biological activities, and SOA produced from these VOCs are positively correlated with MSA (Dall'Osto et al., 2012; Willis et al., 2016; Kim et al., 2017; Willis et al., 2017). While the mixing ratios of isoprene and monoterpenes are typically quite low over open oceans (Hu et al.,

2013) due to their weak emissions, on rare occasions, elevated monoterpene mixing ratios up to ~100 ppt were observed (Kim et al., 2017), possibly due to enhanced microorganism growth as a result of nutrient replenishment (Kim et al., 2017). The second type of VOCs are produced by the oxidation reactions at the air-sea interface, especially when the sea surface microlayer is enriched in organic surfactants (Mungall et al., 2017; Brüggemann et al., 2018). These water-soluble organics can come from phytoplankton, but can also be from other sources, including other autotrophs and atmospheric depositions

(Wurl et al., 2011). Therefore, this type of oceanic VOCs and thus SOA formed may not correlate with MSA (Wurl et al., 2011; Mungall et al., 2017; Brüggemann et al., 2018).

At present, the contribution of secondary organics to the growth of pre-CCN particles in the MBL and the seasonal variation of this contribution remain unclear, largely due to the scarcity of the pre-CCN particle composition measurements. Existing

studies of pre-CCN growth in the MBL were typically within relatively short time periods (i.e., about 1-month) (Dall'Osto et al., 2012; Willis et al., 2016; Kim et al., 2017; Mungall et al., 2017; Willis et al., 2017; Vaattovaara et al., 2006; Modini et al.,



2009; Bzdek et al., 2014; Lawler et al., 2014; Swan et al., 2016), and were often conducted in coastal regions (Vaattovaara et al., 2006; Modini et al., 2009; Dall'Osto et al., 2012; Bzdek et al., 2014; Lawler et al., 2014; Swan et al., 2016) with substantial influences from continental emissions. Here we present the first long-term observational constraint on the importance of

secondary organics to the growth of pre-CCN particles in remote MBL. Hygroscopicity of size-classified particles was characterized over a period of 14 months in the Eastern North Atlantic. By taking advantage of the contrasting hygroscopicity values of sulfate, MSA, and other secondary organic species, we constrain and identify the major species that are responsible for the growth of pre-CCN particles. Our results show that the organics represent an important or even the dominant condensing species during ~80% of growth events.

**2. Measurements and datasets**

The Eastern North Atlantic (ENA) atmospheric observatory was established by the Atmospheric Radiation Measurement (ARM) Climate Research Facility (https://www.arm.gov/capabilities/observatories/ena) in October 2013. This remote oceanic site, located on Graciosa Island, Azores, Portugal (39° 5' 30" N, 28° 1' 32" W, 30.48 m above mean sea level) (Mather and Voyles, 2013) straddles the boundary between the subtropics and mid-latitudes in the eastern North Atlantic. The ENA is a

region of persistent but diverse marine low clouds, the albedo and precipitation of which are highly susceptible to perturbations of aerosol properties (Wood, 2012; Carslaw et al., 2013). Air masses arriving at this site can originate from North America, Northern Europe, the Arctic, and the Atlantic (Wood et al., 2015; Wang et al., 2016; Zheng et al., 2018). The routine measurements at the ENA site include meteorological parameters, trace gases mixing ratios, and aerosol and cloud properties (Zheng et al., 2018). The relevant routine measurements used in this study are summarized in section 2.3.


From June 2017 to Aug. 2018, the Aerosol and Cloud Experiments in the Eastern North Atlantic (ACE-ENA) campaign (Wang et al., 2016) was conducted in the Azores to investigate the aerosol-cloud interactions in the remote marine boundary layer (MBL). As a key part of this campaign, additional aerosol measurements were carried out at the ENA site, including aerosol size distribution and size-resolved CCN activated fractions (Mei et al., 2013c; Thalman et al., 2017). The instruments and

calibration procedures are detailed elsewhere (Zheng et al., 2020), and are briefly described below. The data from these measurements are available at https://www.arm.gov/research/campaigns/aaf2017ace-ena.

**2.1 Size distribution measurements and mode fittings**

Aerosol size distribution was measured by a scanning mobility particle analyzer (SMPS, Model 3938, TSI Incorporated, Shoreview, MN, USA). Dry (RH < 25%) aerosol number size distribution ranging from 10 to 470 nm in particle diameter was

measured every 8 minutes. In addition, a condensation particle counter (CPC, Model 3772, TSI Incorporated, Shoreview, MN, USA) was operated side-by-side to measure the total aerosol number concentrations (CN) concurrently. The measured aerosol number size distributions are fitted as a sum of up to three lognormal modes. Based on the fitted mode geometric mean





diameters ($D_{p,n}$), the fitted modes are classified as the nucleation mode ($D_{p,n} < 20$ nm), the Aitken mode ($20 < D_{p,n} <\sim 80$ nm), the accumulation mode ($\sim 80 < D_{p,n} < \sim 300$ nm), and the sea spray aerosol mode ($D_{p,n} > \sim 300$ nm) (Quinn et al., 2017; Zheng
et al., 2018).

## 2.2 Size-resolved CCN activated fraction measurements

The size-resolved CCN measurement system (SCCN) consists of a Differential Mobility Analyzer (DMA, TSI Inc., Model 3081) coupled to a CPC (TSI Inc., Model 3010) and a cloud condensation nuclei counter (CCNC, Droplet Measurement Technologies, Boulder, CO) (Frank et al., 2006; Moore et al., 2010; Petters et al., 2007; Mei et al., 2013b). This system
measures the activated fraction (i.e., the fraction of particles that activate and form cloud droplets) of size-classified particles as a function of super-saturation (Thalman et al., 2017). During the ACE-ENA campaign, the DMA stepped through 6 dry particle diameters ($D_{p,\,SCCN}$) of 40, 50, 75, 100, 125, and 150 nm. At each $D_{p,\,SCCN}$, the super-saturation level inside the CCNC was varied by changing the flow rate and/or temperature gradient $\triangle T$. The corresponding supersaturation levels, ranging from 0.07% to 1.34% at 298 K, were calibrated using ammonium sulfate particles following established procedures (Lance et al.,
2013; Mei et al., 2013a; Thalman et al., 2017). An entire measurement cycle through the 6 particle diameters took between 1~2 h, depending on particle number concentration. Temperature dependence of CCNC supersaturation (Rose et al., 2008; Thalman et al., 2017) and the effect of multi-charged particles (Thalman et al., 2017) are taken into consideration. The particle hygroscopicity parameter under supersaturated conditions, $\kappa_{CCN}$ (Petters and Kreidenweis, 2007), is derived from the activated fraction spectrum and the corresponding particle diameter (Lance et al., 2013; Mei et al., 2013a; Thalman et al., 2017).

## 2.3 Other relevant datasets used in this study

Routine measurements at the ENA site used in this study include the non-refractory submicron aerosol (NR-PM$_1$) composition (organics, sulfate, nitrate, ammonium, and chloride) characterized by an Aerosol Chemical Speciation Monitor (ACSM; Aerodyne Research, Inc., Billerica, MA, USA) (Watson, 2017) and particle hygroscopic growth measured by a Humidified Tandem Differential Mobility Analyzer (HTDMA, Brechtel Manufacturing Inc., CA, USA) (Uin, 2016). The HTDMA
measures aerosol hygroscopic growth factor under ~80% RH at 5 particle diameters (50, 100, 150, 200 and 250 nm), from which the aerosol hygroscopicity under sub-saturated conditions ($\kappa_{GF}$) is derived (Petters and Kreidenweis, 2007).

Gas-phase SO$_2$ and MSA concentrations are from the Modern-Era Retrospective Analysis for Research and Applications, version 2 (MERRA-2) reanalysis data (Gelaro et al., 2017), at the grid corresponding to the ENA site.

## 3. Derivation of the hygroscopicity parameter of condensing species

Continuous growth of Aitken mode particles is identified from the aerosol size distribution time series. As a result of the condensational growth, aerosol chemical compositions and thus the hygroscopicity of Aitken mode particles are expected to



evolve with time during the growth events. As potential condensing species (Table S1) have contrasting hygroscopicity parameters, the variation of hygroscopicity parameter during the growth events can therefore be used to infer the major

condensing species.

### 3.1 Matching aerosol size modes with the hygroscopicity measurements

Here we detail the procedure to correlate the aerosol size distribution with the SCCN measurements. The same procedure is also applied to correlate the aerosol size distribution with the HTDMA measurements. The CCN activated fraction spectrum was measured at 6 fixed sizes ($D_{p, SCCN}$). As the size of Aitken mode particles evolves continuously during the growth events,

we first determine if the hygroscopicity of the growing Aitken mode can be captured by the measurement at one of the six $D_{p, SCCN}$ using the following two criteria. The first criterion is that $D_{p, SCCN}$ (e.g., 40 nm, 50nm, or 75 nm) is within one geometric standard deviation ($\sigma$) of the Aitken mode diameter, i.e., $D_{p, n} \sigma^{-1} < D_{p, SCCN} < D_{p, n} \sigma$ (Fig. 1a). For example, at time $t_0$, $D_{p, SCCN}$ (40 nm) is within one $\sigma$ range of the Aitken mode diameter (i.e., dark blue shaded area in Fig. 1a), and the $\kappa_{CCN}$ value measured at 40 nm is considered representative of the Aitken mode (solid blue curve in Fig. 1a). In contrast, at a later time $t_1$, the Aitken

mode grew to larger sizes (dash blue curve in Fig. 1a), and 40 nm became smaller than $D_{p, n} \sigma^{-1}$ (light blue shaded area in Fig. 1a). Therefore, $\kappa_{CCN}$ measured at 40 nm no longer represents the hygroscopicity of the Aitken mode at $t_1$. The second criterion is that particle concentration at $D_{p, SCCN}$ is dominated by the Aitken mode only (Fig. 1b), i.e., over 95% of the particles at the measured $D_{p, SCCN}$ is contributed by the Aitken mode. As an example, both the Aitken mode (blue curve) and the accumulation mode (red curve) contribute to the number size distribution at $D_{p, SCCN}$ (black dash line, Fig. 1b). Although $D_{p, SCCN}$ is within

one $\sigma$ of the Aitken mode diameter, the contribution of Aitken mode is less than 95% at this size (orange curve in Fig. 1b). Therefore, measurement at $D_{p, SCCN}$ is not deemed as representative of the Aitken mode due to the substantial contribution from accumulation mode particles. Only data points that meet both criteria are selected, as illustrated in Fig. 1c. Figure 2a gives an example of the time series of Aitken mode diameter and paired $\kappa_{CCN}$ value during a growth event.

### 3.2 Derivation of the hygroscopicity of condensed species ($\kappa_c$) during growth events

The derivation is applied to condensational growth events when there are sufficient number (> 6 points) of $\kappa$ measurements that satisfy both criteria described in M2.1. For each condensational growth event selected (e.g., Fig. 2a), the average hygroscopicity parameter of condensing species, $\kappa_c$, is derived based on the following three assumptions. Here, $\kappa$ represents either the hygroscopicity derived from SCCN (i.e., $\kappa_{CCN}$) or HTDMA (i.e., $\kappa_{GF}$) data.

The first assumption is that the change in particle volume (diameter) is due to the condensational growth only, namely:
$$V_c = \Delta V = V_1 - V_0 = (\pi/6) D_{p1}^3 - (\pi/6) D_{p0}^3 \qquad (1)$$
where $V$ is the particle volume and $D_p$ is the particle diameter. Hereinafter we use $X_1$ and $X_0$ to denote the corresponding particle property $X$ after and before the condensational growth, respectively, and $X_c$ refers to the property $X$ of the condensed species. The second assumption is that the aerosol $\kappa$ follows the volume-weighted mixing law (Petters and Kreidenweis, 2007):





$$\kappa_1 = \kappa_0 (V_0 / V_1) + \kappa_c (V_c / V_1) \qquad (2)$$

The third assumption is that the growth rate is identical for particles of the same size, and thus the relative position of any given particle in the accumulative size distribution is maintained throughout the growth. Let $CDF_0$ and $CDF_1$ denote the particle cumulative size distributions before and after the particle growth, and $D_{p0}$ and $D_{p1}$ represent particle diameters before and after particle growth, respectively. The number of particles smaller than $D_{p1}$ following particle growth should be the same as the

number of particles smaller than $D_{p0}$ prior to the growth event (Fig. 2b):

$$CDF_1(D_{p1}) = CDF_0(D_{p0}) \qquad (3)$$

For each particle size (i.e., $D_{p1}$) measured during the growth events, the original particle size (i.e., $D_{p0}$) is derived from Eq. (3). The volume fraction of condensed species, $f_{V, cond}$, is given by:

$$f_{V, cond} = V_c/V_1 = 1 - V_0/V_1 = 1 - (D_{p0} / D_{p1})^3 \qquad (4)$$

By combining Eq. 1-4, we have:

$$\kappa_1 = (\kappa_c - \kappa_0) f_{V, cond} + \kappa_0 \qquad (5)$$

Both $\kappa_1$ and $f_{V, cond}$ are from the measurements as described above. Therefore, $\kappa_c$ and $\kappa_0$ can be derived from the linear fitting of $\kappa_1$ vs. $f_{V, cond}$ for each growth event (e.g., Fig. 2c), where $\kappa_0$ is the intercept, and $\kappa_c$ is the sum of slope and intercept. The method described here was applied to both SCCN and HTDMA measurements, and $\kappa_c$ derived are referred to as $\kappa_{c,CCN}$ and $\kappa_{c,GF}$

hereinafter, respectively.

## 4. Constraining the major condensing species in remote MBL

Figure 3 shows two examples of the identified growth events, with the dominant condensing species being sulfate and organics, respectively. While the measured $\kappa_{CCN}$ of the Aitken mode particles (i.e., pre-CCN particles that are below ~ 80 nm) are similar (~0.45) at the start of both events, the variations of $\kappa_{CCN}$ with growing particle size show opposite trends. For the July case

(Fig. 3a,b), $\kappa_{CCN}$ increased with the volume fraction of condensed species ($f_{V,cond}$, Fig. 3b), indicating that the hygroscopicity of the condensed species, $\kappa_{c,CCN}$, exceeds that of the original particles. The derived $\kappa_{c,CCN}$ value is 0.7, which is typical of sulfates (Table S1). In contrast, during the September growth event (Fig. 3c,d), $\kappa_{CCN}$ decreased as the particles grew. The derived $\kappa_{c,CCN}$ value is ~0.3, indicating organics as the dominant condensing species. We note that $\kappa_{c,CCN}$ is derived from the volume-weighted mixing law (Petters and Kreidenweis, 2007) (i.e., ideal Zdanovskii, Stokes, and Robinson (ZSR) mixing).

Organic surfactants may facilitate CCN activation by lowering surface tension of growing droplets (Ovadnevaite et al., 2017). In scenarios when particles contain organic surfactants, particle hygroscopicity $\kappa_{CCN}$ may be greater than the simple volume average of participating species. As a result, the derived $\kappa_{c,CCN}$ value based on the volume-weighted mixing law may be overestimated, therefore leading to an underestimation of the contribution of organics to the particle condensational growth.

A total of 62 growth events are identified during the 14-month campaign (Fig. 4). These events are classified into 3 categories according to the derived $\kappa_{c,CCN}$ value (Table S1): (1) low hygroscopicity (i.e., $\kappa_{c,CCN} < 0.45$) indicating organics as the dominant





condensing species, (2) high hygroscopicity (i.e., $\kappa_{c,CCN} > 0.65$) with acidic sulfate (i.e., $H_2SO_4$ or $NH_4HSO_4$) dominating the particle condensational growth, and (3) intermediate hygroscopicity value (i.e., $0.45 < \kappa_{c,CCN} < 0.65$), when $(NH_4)_2SO_4$ and/or mixtures of organics and acidic sulfate contribute to the particle growth.

**5. Monthly distributions of the dominant condensing species**

The monthly distribution of the identified growth events and the dominant condensing species are shown in Fig. 4. Relatively more events were observed during the summer seasons due to favorable synoptic conditions. In summer, there is a stronger influence by the Azores High while the influence from mid-latitude cyclones and the corresponding wet scavenging are much weaker (Zheng et al., 2018). The distribution of the event categories shows that, contrary to the conventional thinking,

$NH_4HSO_4/H_2SO_4$ dominated the condensational growth during only 18% of the growth events. This is less than the events dominated by organics at 24%. The majority (58%) of the growth events exhibit intermediate $\kappa_{c,CCN}$ values, suggesting that $(NH_4)_2SO_4$ or a mixture of organics and sulfate are responsible for the particle condensational growth.

To further constrain the condensing species for the intermediate $\kappa_{c,CCN}$ category, we compare the $\kappa_{c,CCN}$ value with the

hygroscopicity under sub-saturated conditions ($\kappa_{c,GF}$), which is derived from measured particle hygroscopic growth (section 3). For $(NH_4)_2SO_4$, the difference between $\kappa_{c,CCN}$ and $\kappa_{c,GF}$ is relatively small (within 20%) (Petters and Kreidenweis, 2007), while the difference is usually substantially larger (Wex et al., 2009; Rastak et al., 2017; Petters et al., 2009; Pajunoja et al., 2015; Ovadnevaite et al., 2011; Massoli et al., 2010) for organic species. The large difference has been attributed to the solution non-ideality (Petters et al., 2009), the formation of hydrogels (Ovadnevaite et al., 2011), and the solubility and phase states

(Pajunoja et al., 2015; Rastak et al., 2017). One example of the intermediate $\kappa_{c,CCN}$ category is shown in Fig. S1. For this case, the derived $\kappa_{c,CCN}$ and $\kappa_{c,GF}$ values are 0.59 and 0.45, respectively (Fig. S1). The difference is close to the measurement uncertainty (i.e., 20%), and therefore the major condensing species for this example is classified as $(NH_4)_2SO_4$.

Figure 5 compares the values of $\kappa_{c,CCN}$ and $\kappa_{c,GF}$ for all available events in the intermediate $\kappa_{c,CCN}$ category. For most of these

events, $\kappa_{c,GF}$ is at least 20% lower than $\kappa_{c,CCN}$, indicating organics likely played an important role in particle condensational growth. In addition, chemical composition of sub-micron non-refractory aerosol (NR-PM$_1$; aerodynamic diameters below 1 μm) indicates an ammonium-poor condition over the ENA (color bar in Fig. 5), typical of remote marine environment (Adams et al., 1999). Therefore, sulfate is not fully neutralized as $(NH_4)_2SO_4$. These evidences suggest that during most of the intermediate- $\kappa_{c,CCN}$ events, the condensed species are a mixture of sulfates and organics instead of dominated by $(NH_4)_2SO_4$.

Based on a $\kappa_{CCN}$ value of 0.9 for acidic sulfates ($H_2SO_4$ and/or $NH_4HSO_4$, Table S1), the average contribution of organics during the intermediate-$\kappa_{c,CCN}$ events ranges from 42% and 63%, depending on the $\kappa_{CCN}$ values of organics assumed (0.1~0.36; Table S1). Therefore, organics played an important role during the intermediate-$\kappa_{c,CCN}$ events and dominated the particle



condensational growth for the low-$\kappa_{c,CCN}$ category. Together, these two categories represent a total of ~80% of the growth events and occurred throughout the year.

**6. Sources of the condensing organics**

Given the importance of secondary organics to particle condensational growth, the potential sources of the condensing organics are investigated by examining the air mass origins (SI S1). Here we classify the origin of air mass during the growth events into four types: (1) continental air masses from North America or Europe, (2) the Arctic, (3) the subtropical, and (4) the mid-latitude Atlantic. Note that an air mass is denoted as continental if it passed over the North America or Europe, so the non-continental types represent the air masses that had stayed over oceans or clean continental areas (i.e., Arctic region) for at least 10 days (SI S1).

Growth events of mid-latitude Atlantic or Arctic type were observed exclusively from May to September, a period that coincides with the phytoplankton blooms in mid-latitude Atlantic or Arctic, but not the subtropics (Sapiano et al., 2012). For these events, $\kappa_{c,CCN}$ is anti-correlated with MSA/SO$_2$ ratio (Fig. 6a), which is from MERRA-2 reanalysis data (section 2.3). As fixed yields of SO$_2$ and MSA from DMS oxidation are assumed in MERRA-2 data (Chin et al., 2000; Randles et al., 2017), a lower MSA/SO$_2$ ratio suggests other SO$_2$ sources in addition to DMS oxidation contribute to these events. These other sources could include volcanic emissions and combustion products from international shipping (Randles et al., 2017). As MSA is a tracer of biogenically derived SOA in marine environment (Seinfeld and Pandis, 2016), the anti-correlation also indicates that the condensed organics are likely SOA produced from VOCs emitted from ocean biological activities (e.g., phytoplankton blooms). The value of $\kappa_{c,CCN}$ is not correlated with the NR-PM$_1$ organic/sulfate ratio (Fig. 6b), suggesting different sources of the condensed species in pre-CCN and the accumulation mode particle composition.

Among the remaining growth events, only four of them are subtropical cases, which occurred outside the bloom periods. During the other events, air masses were potentially influenced by continental emissions (Fig. S2). For these events, $\kappa_{c,CCN}$ is instead positively correlated with MSA/SO$_2$ ratio (Fig. 6c), indicating that secondary organics formed from phytoplankton-emitted VOCs likely played a minor role in the observed particle condensational growth. The $\kappa_{c,CCN}$ value generally decreases with increasing NR-PM$_1$ organic / sulfate ratios (Fig. 6d), suggesting that the formation of SOA led to increased organic fraction for both pre-CCN and accumulation mode particles. Possible sources of the condensed organics include SOA generated from long-range transported continental VOCs and VOCs released by the sea-surface microlayer oxidation that are not directly related to phytoplankton emissions.

As continentally emitted VOCs are removed by oxidation during long-range transport, it is expected that in-situ SOA production from these VOCs is low and plays a minor role in particle condensational growth over the remote oceans (Kelly et

al., 2019; D'Andrea et al., 2013). On the other hand, aromatic compounds were detected in pre-CCN particles in clean air masses at a coastal site (Lawler et al., 2014), indicating potential contribution of SOA from anthropogenic VOCs with long lifetime. Oxidation reactions at the air-sea interface can produce VOCs, which lead to subsequent SOA formation (Mungall et al., 2017; Brüggemann et al., 2018). This VOC source is present all-year round, even during winter when there is little biological activity in the ocean (Brüggemann et al., 2018). Therefore, the secondary organics produced via this pathway can

contribute to the growth of pre-CCN particles outside the biologically active seasons of the ocean.

## 7. Conclusions

In summary, we show that during all seasons, secondary organics play an important role in the condensational growth of pre-CCN particles, and by extension, the formation of CCN in the remote marine boundary layer. The secondary organic species likely derive from a variety of precursors, including VOCs produced from marine biogenic activity, continentally emitted

VOCs with long lifetime that survive the long-range transport, and VOCs formed by oxidation at the air-sea interface. Current global models typically assume that sulfates dominate the particle growth over remote oceans, and therefore may substantially underestimate the formation of CCN by condensational growth in remote marine boundary layer.


*Data availability.* All data used in this study are available at https://www.arm.gov/research/campaigns/aaf2017ace-ena and https://www-air.larc.nasa.gov/missions/naames/index.html.

*Author contributions.* J.W. and G.Z. designed the study. J.W. G.Z., C.K., J.U., and T.W carried out the measurements, G.Z.
and J.W. conducted the analysis and wrote the manuscript with contributions from all authors.

*Competing interests.* The authors declare no competing interests.

*Acknowledgments.* The research was conducted with funding from the Atmospheric System Research (ASR) program (Award
No. DE-SC0020259), Office of Biological and Environmental Research (OBER) of the United States Department of Energy. We acknowledge additional support by the Atmospheric Radiation Measurement (ARM) Climate Research Facility, a user facility of the United States Department of Energy, Office of Science, sponsored by the Office of Biological and Environmental Research.



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



**Figures**

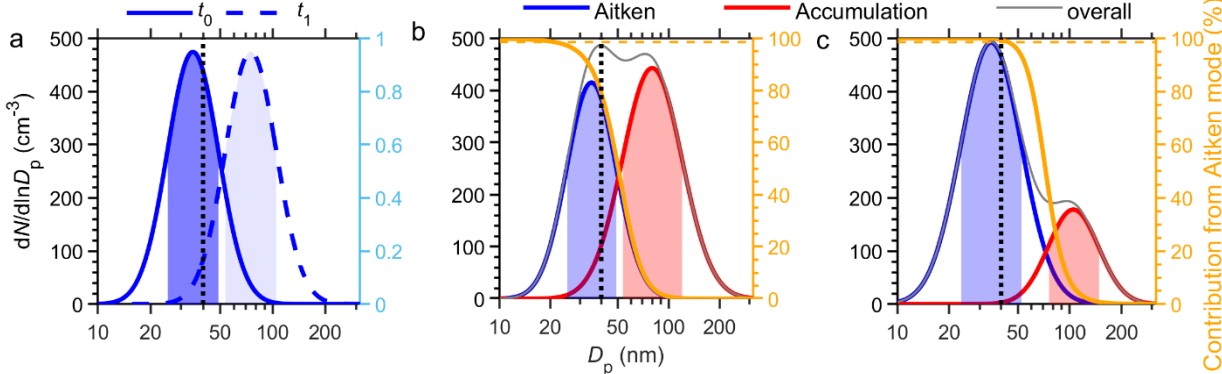

**Figure 1. Matching aerosol Aitken mode with hygroscopicity measurements at fixed particle diameters.** The black dash lines indicate the selected particle size (i.e., $D_{p,\,SCCN}$) at which the hygroscopicity parameter $\kappa$ is derived. The shaded areas indicate one $\sigma$ range from the fitted lognormal mode diameter, $D_{p,\,n}$.




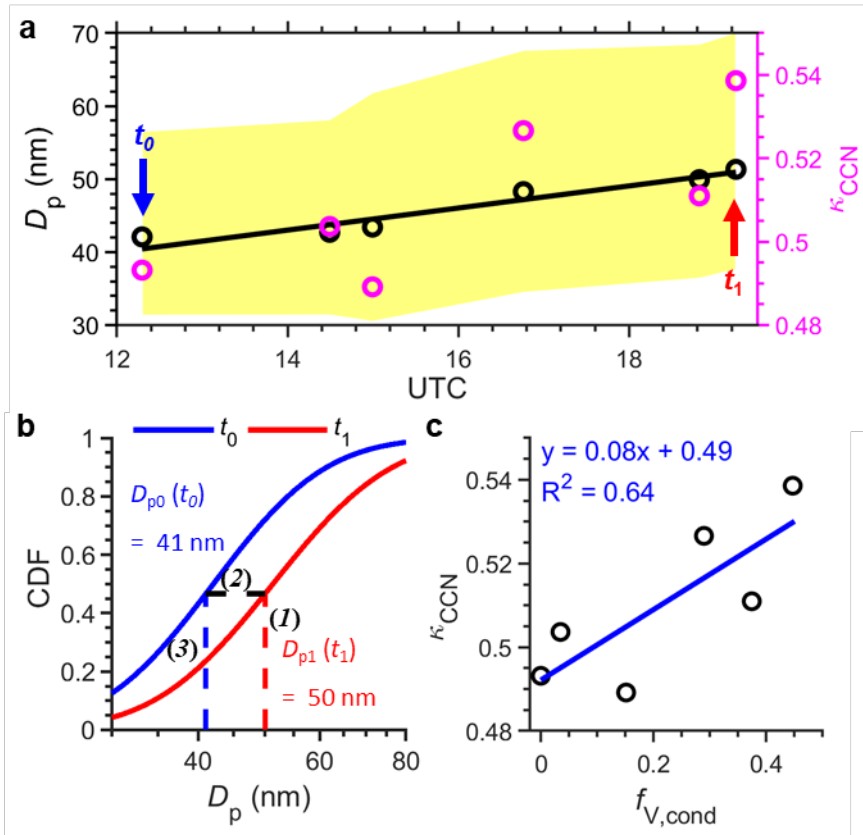

**Figure 2. Derivation of $\kappa_{c,CCN}$ from size-resolved CCN measurement during an example condensational growth event.** (a) Mode diameter and hygroscopicity $\kappa_{CCN}$ of the growing Aitken mode. The black circles are fitted mode diameter, $D_{p,n}$, and the shaded area indicate the one $\sigma$ range of the fitted mode. Black line shows the increasing trend of $D_{p,n}$, which is used to identify growth events. (b) Derivation of the original particle dimeter ($D_{p0}$) at the beginning of the growth event from particle diameter after growth ($D_{p1}$) using cumulative particle size distributions. (c) Derivation of $\kappa_{c,CCN}$ through linear fitting of $\kappa_{CCN}$ versus $f_{V,cond}$.





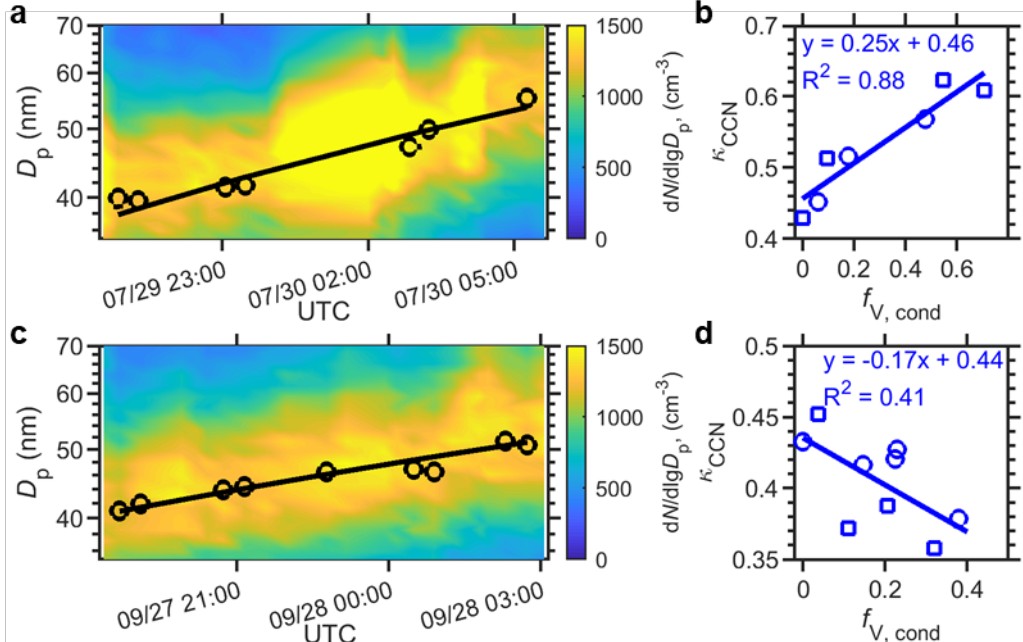

**Figure 3. Examples of pre-CCN particle growth dominated by (a,b) acidic sulfates and (c,d) organics, respectively, as observed in 2017.** (a)(c) Examples of growth events identified from the time series of measured aerosol size distribution. The black circles indicate lognormal-fitted Aitken mode diameter, and the black lines indicate the growth of the mode diameter (see section 3). (b)(d) Particle hygroscopicity $\kappa_{CCN}$ as a function of the volume fraction of condensed species in the growing particles ($f_{V,cond}$). $f_{V,cond}$ increases as particles grow by condensation. The value of $\kappa_{c,CCN}$ is derived from the variation of $\kappa_{CCN}$ with $f_{V,cond}$ (section 3).






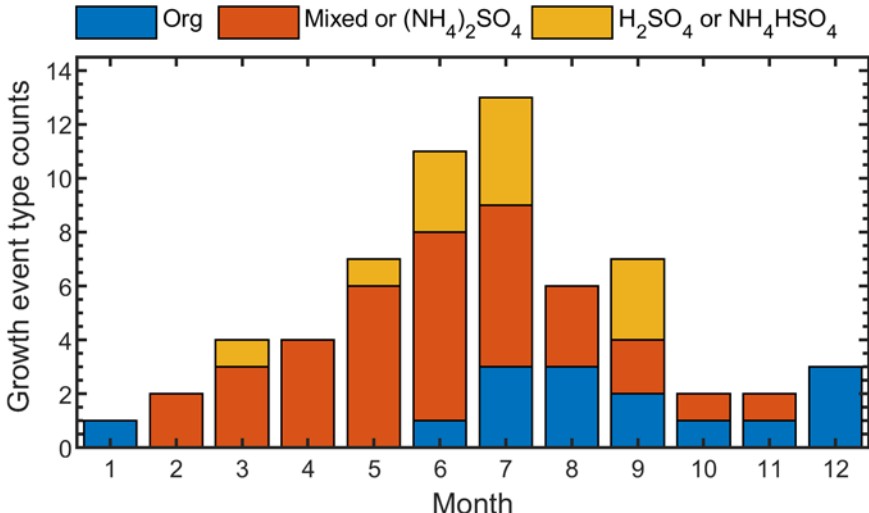

**Figure 4. Monthly distribution of observed condensational growth events and the category of dominant condensing species during the ACE-ENA campaign.**






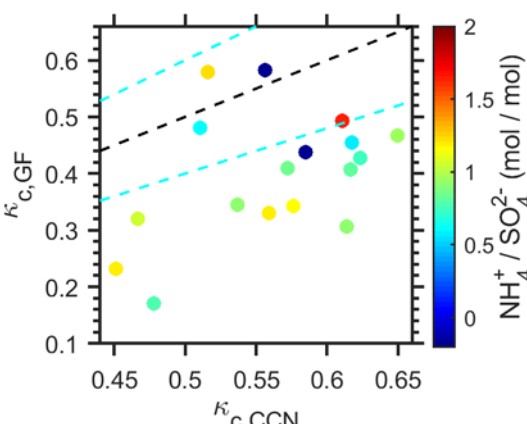

**Figure 5. Comparison of $\kappa_{c,CCN}$ and $\kappa_{c,GF}$ values for the intermediate $\kappa_{c,CCN}$ category, colored by the measured molar ratios of NH$_4^+$/SO$_4^{2-}$.**


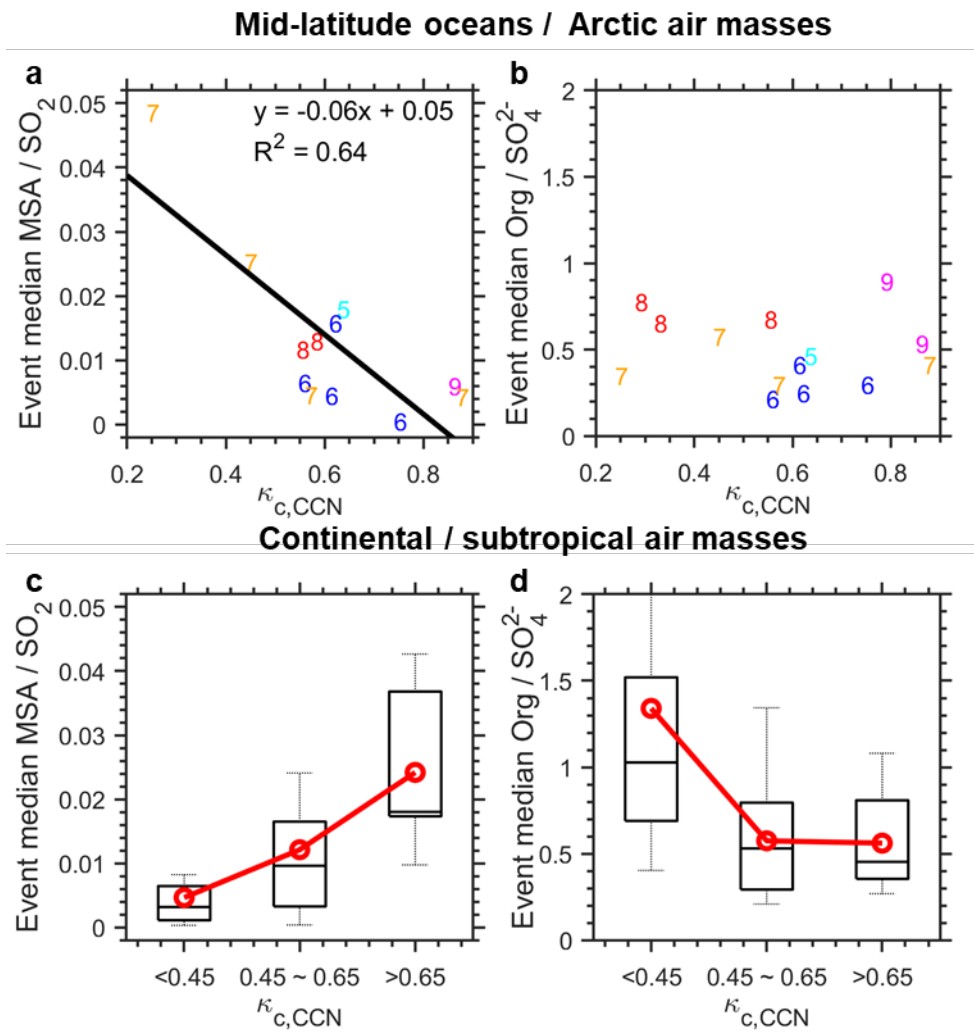

**Figure 6. Potential sources of the condensing organics.** Correlations of the derived $\kappa_{c,CCN}$ with (a,c) MSA/SO$_2$ ratio and (b,d) non-refractory PM$_1$ organics/SO$_4^{2-}$ ratio, for (a,b) the clean air masses from mid-latitude Atlantic or Arctic, and (c,d) the continental or subtropical air masses. Numbers shown in (a, b) indicate the month in which the growth events occurred.