# Peer review of "Large contribution of organics to condensational growth and formation of cloud condensation nuclei (CCN) in remote marine boundary layer"

_Atmospheric Chemistry and Physics, 2020_

## Referee Comment (RC1) · Anonymous Referee #1 · 8 Jul 2020

Zheng et al present a long time series of CCN measurements in the remote marine boundary layer. Measurements of this kind are rare, especially during nucleation events. The authors systematically characterize a series of nucleation events to calculate the hygroscopicity parameter for the condensing material. Surprisingly, they find that it is much lower than that for sulfate as would be predicted. These measurements provide some of the first direct evidence that condensation of organic material in the marine boundary layer contributes to particle growth with important implications for CCN. The paper is well written and should be published in ACP. I have only a few comments:

[Figure]

1) Are there any direct gas-phase measurements that can be used to support the air-mass characterization efforts? Even CO would be helpful!

2) It would be helpful if the authors could include some context for what the kappa value is for different potential condensing species (e.g., BVOC oxidation products, MSA, carbonyl compounds which may dominate the photo-chemical VOC pathway).

3) Is the MSA-SO2 yield really constant in MERRA-2 or does it still depend on temperature as that dictates the branching between H-abstraction and OH addition in DMS+OH? This is what yields distinct MSA/SO2 ratios. Is it possible to simply use the concentration of SO2 as an indicator?

4) Is it possible to extract from the growth rates any information on the concentration of condensing species? This would help in the comparison with the magnitude of BVOC ocean emissions (e.g., is 10ppt steady-state monoterpene sufficient to sustain this type of growth?)

---

## Referee Comment (RC2) · Anonymous Referee #2 · 21 Jul 2020

Through size-resolved CCN and HTDMA measurements the authors present evidence for a substantial role of organics in the condensational growth of particles to CCN sizes in the remote marine boundary layer. There is no shortage of aerosol organics in the marine atmosphere but there is a lack of information about their sources and impacts. This paper provides information about the role they may play in CCN formation. The paper should be published after the concerns listed below have been addressed.

Line 34: "It has long been recognized. . ." adding a few references going further back than 2018 would be appropriate.

Lines 112 – 113: What is the uncertainty associated with the SO2 and MSA concentrations derived from MERRA-2?

Lines 201 – 202 and Figure S1 caption: These seem contradictory. The main text says "The difference is close to the measurements uncertainty. . ..therefore the major condensing species is classified as (NH4)2SO4. The figure caption says "the major condensing species included both organics and sulfates or dominated by (NH4)2SO4".

Figure 5 caption: Please describe what the black dashed line in the figure represents.

Figure 5: There is no clear relationship between the degree of difference between kappa_c,GF and kappa_c,CCN and the NH4 to SO4 molar ratio. If I'm interpreting the figure correctly, there are instances (dark blue points) when the NH4 to SO4 molar ratio is very low but the difference between the kappa values is small. Based on these data, it's not clear that low amounts of NH4 relative to SO4 is most prevalent during intermediate kappa_c,CCN events. Maybe it would be clearer if Figure 5 were expanded to include all data, not just intermediate events.

Lines 231 – 232: It is stated that kappa_c,CCN is not correlated with the NR-PM1 organic/sulfate ratio suggesting different sources of the condensed species in pre-CCN and the accumulation mode particle composition. Does this lack of a correlation suggest anything about the importance of the pre-CCN condensed species in terms of CCN activity or concentration since the accumulation mode can dominate the CCN concentration?

---

## Referee Comment (RC3) · Anonymous Referee #3 · 25 Jul 2020

Zheng et al. reported long-term measurements of hygroscopicity and composition of pre-CCN particles in a remote marine boundary layer site. They found that for most of the particle condensational growth events, the dominant condensing species are organics instead of sulfate. This paper is well written. I recommend publishing the paper after some minor revisions:

1. Given that there were in-situ ACSM measurements, the authors may be able to separate MSA from the rest of the organics in the ACSM spectra (m/z 79) to better quantify the contribution of MSA and non-DMS VOC to aerosol growth. (Ref: Hodshire et al., The potential role of methanesulfonic acid (MSA) in aerosol formation and growth

and the associated radiative forcings, 2019 ACP, Supporting Information)

2. Line 61: It is unclear in the manuscript what the kappa value is for MSA.

3. Some figure captions in the manuscript are very short. I suggest that the authors include more descriptive information in figure captions. For example, what do the blue dotted lines in Figure 5 represent?

---

## Author Comment (AC1) · 31 Aug 2020

**Manuscript No.**: acp-2020-625

**Title**: Large contribution of organics to condensational growth and formation of cloud condensation nuclei (CCN) in remote marine boundary layer

5   We thank the anonymous referees for their valuable and constructive comments/suggestions on our manuscript. We have revised the manuscript accordingly and please find our point-to-point responses below.

**Comments by Anonymous Referee #1:**

10   *General Comments:*

*Zheng et al present a long time series of CCN measurements in the remote marine boundary layer. Measurements of this kind are rare, especially during nucleation events. The authors systematically characterize a series of nucleation events to calculate the hygroscopicity parameter for the condensing material. Surprisingly, they find that it is much lower than that for sulfate as would be predicted. These*
15   *measurements provide some of the first direct evidence that condensation of organic material in the marine boundary layer contributes to particle growth with important implications for CCN. The paper is well written and should be published in ACP. I have only a few comments:*

*Detailed Comments:*

20   *1) Are there any direct gas-phase measurements that can be used to support the airmass characterization efforts? Even CO would be helpful!*

**Responses:**

Long-term trace gas measurements at the ENA site include CO and $O_3$, both of which come mainly from long-range transport (Zheng et al., 2018). Following the reviewer's suggestion, we compared the mixing
25   ratios of CO and $O_3$ of different air mass origins (Fig. R1). Both CO and $O_3$ mixing ratios are the highest in continental air masses and lowest for mid-latitude ocean air masses, supporting the airmass classification.

We've added this information into the modified manuscript (Line 229-230 in the revised manuscript):

"Here we classify the origin of air mass during the growth events into four types: (1) continental air masses from North America or Europe, (2) the Arctic, (3) the subtropical, and (4) the mid-latitude Atlantic. … The mixing ratios of CO and $O_3$ measured at the ENA site exhibit the highest and lower values for the continental and mid-latitude ocean airmasses, respectively (Fig. S3), supporting the effectiveness of the classifications."

And Fig. R1 is added as Fig. S3 in modified SI.

[Figure]

**Fig. R1** *(added as Figure S3 in updated SI)* **Trace gas mixing ratios in different air mass origins.** (a) CO and (b) $O_3$.

*2) It would be helpful if the authors could include some context for what the kappa value is for different potential condensing species (e.g., BVOC oxidation products, MSA, carbonyl compounds which may dominate the photo-chemical VOC pathway).*

**Responses:**

Previously we've listed this information in Table S1. To make the relevant information clearer, we've moved Table S1 into the modified manuscript as Table 1.

**Table R1** *(added as Table 1 in updated manuscript)***: Hygroscopicity parameter $\kappa$ of potential condensing species over remote oceans**

| Compound | $\kappa_{GF}$ | $\kappa_{CCN}$ | Reference |
|---|---|---|---|

| | | | |
|---|---|---|---|
| $H_2SO_4$ | 1.19 | 0.90 | (Petters and Kreidenweis, 2007) |
| $NH_4HSO_4$ | 1.0 | 0.9 | (Schmale et al., 2018) |
| $(NH_4)_3H(SO_4)_2$ | 0.51 | 0.65 | (Petters and Kreidenweis, 2007) |
| $(NH_4)_2SO_4$ | 0.53 | 0.61 | (Petters and Kreidenweis, 2007) |
| $CH_3SO_3H$ (MSA) | 0.36 | <0.44 | (Johnson et al., 2004; Tang et al., 2019) |
| $\alpha-$pinene/$O_3$/dark SOA | 0.022~0.037 | 0.1 | (Petters and Kreidenweis, 2007) |
| $\beta-$pinene/$O_3$/dark SOA | 0.009~0.022 | 0.1 | (Petters and Kreidenweis, 2007) |
| SOA particles generated via OH radical oxidation | 0~0.3 (20% to 50% lower than corresponding $\kappa_{CCN}$) | 0~0.3 (Generally below the line of: (0.29 ± 0.05)*O:C | (Chang et al., 2010; Massoli et al., 2010) |

*3) Is the MSA-SO2 yield really constant in MERRA-2 or does it still depend on temperature as that dictates the branching between H-abstraction and OH addition in DMS+OH? This is what yields distinct MSA/SO2*
5   *ratios. Is it possible to simply use the concentration of SO2 as an indicator?*

**Responses:**

As pointed out by the reviewer, the DMS branching ratio depends on many factors, including the temperature (Arsene et al., 2001; Hynes et al., 1986; Williams et al., 2001; Yin et al., 1990). However, to the best of our knowledge, a constant DMS branching ratio ($SO_2$:MSA = 75:25) is employed in MERRA-2
10   for simplicity (Chin et al., 2000; Randles et al., 2017), as in the standard GEOS-Chem model (Chatfield and Crutzen, 1990; Chin et al., 1996). Besides the branching ratio, several factors also contribute to the variation of the MSA/$SO_2$ concentration ratio, including other sources of $SO_2$ and the sinks of MSA and $SO_2$.  We note that the MSA/$SO_2$ concentration ratio is substantially lower than any reported branching ratio, and one likely explanation is that $SO_2$ from other sources, including long range transported
15   continental emissions, and emissions from volcanos and shipping, contributes substantially to the variation of MSA/$SO_2$ concentration ratio.

The average hygroscopicity of the condensed species, $\kappa_c$, is expected to strongly correlate with the volume ratios of condensed organics and inorganics (e.g., sulfate), instead of their absolute concentrations. In this study, MSA/$SO_2$ ratio is used as a surrogate of the relative abundance of condensing biogenic secondary

organics and sulfates, which is expected to anti-correlate with $\kappa_c$. We don't expect the concentration of $SO_2$ as an effective indictor, as $SO_2$ concentration alone does not correlate well with the volume ratio of condensed organics and inorganics, and thus $\kappa_c$.

5   *4) Is it possible to extract from the growth rates any information on the concentration of condensing species? This would help in the comparison with the magnitude of BVOC ocean emissions (e.g., is 10ppt steady-state monoterpene sufficient to sustain this type of growth?)*

**Responses:**

We thank the reviewer for the suggestion. Extracting the concentration of condensing species and precursor
10  BVOC requires a number of parameters, including the concentrations of oxidants, existing aerosol Fuchs surface area, the volatility of the condensing organics, and the activity coefficients of the organics in the growing particles. A systematic study on the concentration of condensing species and precursors is beyond the scope of this manuscript, but is the focus of our future work.

15

**References**

Arsene, C., Barnes, I., Becker, K. H., and Mocanu, R.: FT-IR product study on the photo-oxidation of dimethyl sulphide in the presence of NOx—temperature dependence, Atmospheric Environment, 35, 3769-3780, 2001.

5    Chang, R. Y. W., Slowik, J. G., Shantz, N. C., Vlasenko, A., Liggio, J., Sjostedt, S. J., Leaitch, W. R., and Abbatt, J. P. D.: The hygroscopicity parameter (κ) of ambient organic aerosol at a field site subject to biogenic and anthropogenic influences: relationship to degree of aerosol oxidation, Atmos. Chem. Phys., 10, 5047-5064, 2010.

Chatfield, R. B. and Crutzen, P. J.: Are there interactions of iodine and sulfur species in marine air 10    photochemistry?, Journal of Geophysical Research: Atmospheres, 95, 22319-22341, 1990.

Chin, M., Jacob, D. J., Gardner, G. M., Foreman-Fowler, M. S., Spiro, P. A., and Savoie, D. L.: A global three-dimensional model of tropospheric sulfate, Journal of Geophysical Research: Atmospheres, 101, 18667-18690, 1996.

Chin, M., Rood, R. B., Lin, S.-J., Müller, J.-F., and Thompson, A. M.: Atmospheric sulfur cycle simulated 15    in the global model GOCART: Model description and global properties, Journal of Geophysical Research: Atmospheres, 105, 24671-24687, 2000.

Hynes, A. J., Wine, P. H., and Semmes, D. H.: Kinetics and mechanism of hydroxyl reactions with organic sulfides, The Journal of Physical Chemistry, 90, 4148-4156, 1986.

Johnson, G. R., Ristovski, Z., and Morawska, L.: Method for measuring the hygroscopic behaviour of 20    lower volatility fractions in an internally mixed aerosol, Journal of Aerosol Science, 35, 443-455, 2004.

Massoli, P., Lambe, A. T., Ahern, A. T., Williams, L. R., Ehn, M., Mikkilä, J., Canagaratna, M. R., Brune, W. H., Onasch, T. B., Jayne, J. T., Petäjä, T., Kulmala, M., Laaksonen, A., Kolb, C. E., Davidovits, P., and Worsnop, D. R.: Relationship between aerosol oxidation level and hygroscopic properties of laboratory generated secondary organic aerosol (SOA) particles, Geophysical Research Letters, 37, 2010.

25    Petters, M. D. and Kreidenweis, S. M.: A single parameter representation of hygroscopic growth and cloud condensation nucleus activity, Atmos. Chem. Phys., 7, 1961-1971, 2007.

Randles, C., Da Silva, A., Buchard, V., Colarco, P., Darmenov, A., Govindaraju, R., Smirnov, A., Holben, B., Ferrare, R., and Hair, J.: The MERRA-2 aerosol reanalysis, 1980 onward. Part I: System description and data assimilation evaluation, Journal of Climate, 30, 6823-6850, 2017.

Schmale, J., Henning, S., Decesari, S., Henzing, B., Keskinen, H., Sellegri, K., Ovadnevaite, J., Pöhlker, M. L., Brito, J., Bougiatioti, A., Kristensson, A., Kalivitis, N., Stavroulas, I., Carbone, S., Jefferson, A., Park, M., Schlag, P., Iwamoto, Y., Aalto, P., Äijälä, M., Bukowiecki, N., Ehn, M., Frank, G., Fröhlich, R., Frumau, A., Herrmann, E., Herrmann, H., Holzinger, R., Kos, G., Kulmala, M., Mihalopoulos, N., Nenes,

5   A., O'Dowd, C., Petäjä, T., Picard, D., Pöhlker, C., Pöschl, U., Poulain, L., Prévôt, A. S. H., Swietlicki, E., Andreae, M. O., Artaxo, P., Wiedensohler, A., Ogren, J., Matsuki, A., Yum, S. S., Stratmann, F., Baltensperger, U., and Gysel, M.: Long-term cloud condensation nuclei number concentration, particle number size distribution and chemical composition measurements at regionally representative observatories, Atmos. Chem. Phys., 18, 2853-2881, 2018.

10  Tang, M., Guo, L., Bai, Y., Huang, R.-J., Wu, Z., Wang, Z., Zhang, G., Ding, X., Hu, M., and Wang, X.: Impacts of methanesulfonate on the cloud condensation nucleation activity of sea salt aerosol, Atmospheric Environment, 201, 13-17, 2019.

Williams, M. B., Campuzano-Jost, P., Bauer, D., and Hynes, A. J.: Kinetic and mechanistic studies of the OH-initiated oxidation of dimethylsulfide at low temperature – A reevaluation of the rate coefficient and

15  branching ratio, Chem Phys Lett, 344, 61-67, 2001.

Yin, F., Grosjean, D., Flagan, R. C., and Seinfeld, J. H.: Photooxidation of dimethyl sulfide and dimethyl disulfide. II: Mechanism evaluation, Journal of Atmospheric Chemistry, 11, 365-399, 1990.

Zheng, G., Wang, Y., Aiken, A. C., Gallo, F., Jensen, M. P., Kollias, P., Kuang, C., Luke, E., Springston, S., Uin, J., Wood, R., and Wang, J.: Marine boundary layer aerosol in the eastern North Atlantic: seasonal

20  variations and key controlling processes, Atmos. Chem. Phys., 18, 17615-17635, 2018.

---

## Author Comment (AC2) · 31 Aug 2020

**Manuscript No.**: acp-2020-625

**Title**: Large contribution of organics to condensational growth and formation of cloud condensation nuclei (CCN) in remote marine boundary layer

We thank the anonymous referees for their valuable and constructive comments/suggestions on our manuscript. We have revised the manuscript accordingly and please find our point-to-point responses below.

**Comments by Anonymous Referee #2:**

*General Comments:*

*Through size-resolved CCN and HTDMA measurements the authors present evidence for a substantial role of organics in the condensational growth of particles to CCN sizes in the remote marine boundary layer. There is no shortage of aerosol organics in the marine atmosphere but there is a lack of information about their sources and impacts.This paper provides information about the role they may play in CCN formation.*
*The paper should be published after the concerns listed below have been addressed.*

*Detailed Comments:*

*1. Line 34: "It has long been recognized: : :" adding a few references going further back than 2018 would be appropriate.*

**Responses:**

Following the reviewer's suggestion, we've added more references, and the sentence now reads:

"It has long been recognized that sulfate produced from DMS oxidation is a major species for particle condensational growth in the remote marine environment (Charlson et al., 1987; O'Dowd et al., 1999; Pandis et al., 1994; Raes and Van Dingenen, 1992; Sanchez et al., 2018)."

*2. Lines 112 – 113: What is the uncertainty associated with the SO2 and MSA concentrations derived from MERRA-2?*

**Responses:**

As far as we know, currently there's no uncertainty study or comparison with observations for MERRA-2
$SO_2$ and MSA concentrations over the open oceans of Eastern North Atlantic. Over the East Asia where $SO_2$ is mainly from anthropogenic emissions, monthly averaged surface $SO_2$ concentrations from MERRA- were compared to measurements at 46 sites of The Acid Deposition Monitoring Network in East Asia (EANET) during the period from 2001 to 2008 (Randles et al., 2016). The relative biases (the ratio of MERRA-2 to EANET) range from 0.968 in winter to 1.418 in the fall, with correlation varying between 0.319 to 0.501 (Randles et al., 2016).

Whereas some uncertainties are expected, MERRA-2 likely provides the best estimate of $SO_2$ and DMS concentration in the absence of measurements. We also note that the conclusions of this study are not based on the absolute concentrations. Rather, we use the relative trends of $SO_2$ and DMS concentrations to infer the potential sources, which is expected to be less influenced by the uncertainties in the concentrations.

*3. Lines 201 – 202 and Figure S1 caption: These seem contradictory. The main text says "The difference is close to the measurements uncertainty…. therefore the major condensing species is classified as (NH4)2SO4. The figure caption says "the major condensing species included both organics and sulfates or dominated by (NH4)2SO4".*

**Responses:**

Sorry for the confusion. We've clarified the caption of Fig. S1:

"**Figure S1. An example case of deriving $\kappa_{c,CCN}$ and $\kappa_{c,GF}$ for the same growth event. …** (b) A $\kappa_{c,CCN}$ value of 0.59 (i.e., the sum of slope and intercept) is derived from the linear fitting of $\kappa_{CCN}$ vs. $f_{V,cond}$. This value falls in the intermediate-$\kappa_{c,CCN}$ category . (c) A $\kappa_{c,GF}$ value of 0.45 is derived from the variation of

$\kappa_{GF}$ following the same approach. Major condensing species of this case is determined to be $(NH_4)_2SO_4$ (see detailed discussions in section 5 of the main text)."

*4. Figure 5 caption: Please describe what the black dashed line in the figure represents.*
**Responses:**

Following the reviewer's suggestion, we've added the description and the caption now reads:

"**Figure 5. Comparison of $\kappa_{c,CCN}$ and $\kappa_{c,GF}$ values for the intermediate $\kappa_{c,CCN}$ category, colored by the measured molar ratios of $NH_4^+/SO_4^{2-}$.** The black dash line is the 1:1 line, while the cyan dash lines represent the ± 20% uncertainties."

*5. Figure 5: There is no clear relationship between the degree of difference between kappa_c,GF and kappa_c,CCN and the NH4 to SO4 molar ratio. If I'm interpreting the figure correctly, there are instances (dark blue points) when the NH4 to SO4 molar ratio is very low but the difference between the kappa values is small. Based on these data, it's not clear that low amounts of NH4 relative to SO4 is most*
*prevalent during intermediate kappa_c,CCN events. Maybe it would be clearer if Figure 5 were expanded to include all data, not just intermediate events.*

**Responses:**

We've clarified this point in the modified manuscript as:

"In addition, chemical composition of sub-micron non-refractory aerosol (NR-PM$_1$; aerodynamic diameters
below 1 μm) indicates an ammonium-poor condition over the ENA (color bar in Fig. 5), typical of remote marine environment (Adams et al., 1999). Therefore, sulfate is not fully neutralized as $(NH_4)_2SO_4$. Note that the bulk $NH_4^+/SO_4^{2-}$ molar ratio shown in Fig. 5 is dominated by that of accumulation mode particles, whereas the $\kappa_{c,CCN}$ and $\kappa_{c,GF}$ values are derived from growing Aitken mode particles. Whereas the degrees of neutralization for accumulation and Aitken modes are expected to correlate with each other (e.g., lower
neutralization degrees for both accumulation and Aitken modes under more ammonium poor conditions), it is possible that the neutralization degree may exhibit size dependence under some circumstances. For example, in-cloud formation of $SO_4^{2-}$ influences the neutralization degree of accumulation mode particles only (Seinfeld and Pandis, 2016), possibly leading to a lower degree of neutralization for accumulation mode. This could explain a few data points that exhibit lower degree of neutralization but similar $\kappa_{c,CCN}$ and
$\kappa_{c,GF}$ values."

*6. Lines 231 – 232: It is stated that kappa_c,CCN is not correlated with the NR-PM1 organic/sulfate ratio suggesting different sources of the condensed species in pre-CCN and the accumulation mode particle composition. Does this lack of a correlation suggest anything about the importance of the pre-CCN*
*condensed species in terms of CCN activity or concentration since the accumulation mode can dominate the CCN concentration?*

**Responses:**

We agree that accumulation mode can dominate the CCN concentration. On the other hand, condensational growth of the pre-CCN represents a major source of CCN in marine boundary layer (Sanchez et al., 2018;
Zheng et al., 2018), and this is the focus of this study. In MBL, once pre-CCN particles grow to sufficient size (e.g., Hoppel minimum diameter) and become CCN, their composition continuously evolves, for example, through in cloud production of sulfate and organics. Therefore, it is not surprising that $\kappa_{c,CCN}$ is not well correlated with accumulation or bulk particle composition under certain conditions. However, the lack of correlation does not suggest that the condensation growth of pre-CCN is not an important source of CCN in MBL.

**References**

Adams, P. J., Seinfeld, J. H., and Koch, D. M.: Global concentrations of tropospheric sulfate, nitrate, and ammonium aerosol simulated in a general circulation model, Journal of Geophysical Research: Atmospheres, 104, 13791-13823, 1999.

Charlson, R. J., Lovelock, J. E., Andreae, M. O., and Warren, S. G.: Oceanic phytoplankton, atmospheric sulphur, cloud albedo and climate, Nature, 326, 655-661, 1987.

O'Dowd, C. D., Lowe, J. A., and Smith, M. H.: Observations and modelling of aerosol growth in marine stratocumulus—case study, Atmospheric Environment, 33, 3053-3062, 1999.

Pandis, S. N., Russell, L. M., and Seinfeld, J. H.: The relationship between DMS flux and CCN concentration in remote marine regions, Journal of Geophysical Research: Atmospheres, 99, 16945-16957, 1994.

Raes, F. and Van Dingenen, R.: Simulations of condensation and cloud condensation nuclei from biogenic SO2 in the remote marine boundary layer, Journal of Geophysical Research: Atmospheres, 97, 12901-12912, 1992.

Randles, C., da Silva, A., Buchard, V., Darmenov, A., Colarco, P., Aquila, V., Bian, H., Nowottnick, E., Pan, X., and Smirnov, A.: The MERRA-2 aerosol assimilation, NASA Tech. Rep. Series on Global
Modeling and Data Assimilation, 45, 2016.

Sanchez, K. J., Chen, C.-L., Russell, L. M., Betha, R., Liu, J., Price, D. J., Massoli, P., Ziemba, L. D., Crosbie, E. C., Moore, R. H., Müller, M., Schiller, S. A., Wisthaler, A., Lee, A. K. Y., Quinn, P. K., Bates, T. S., Porter, J., Bell, T. G., Saltzman, E. S., Vaillancourt, R. D., and Behrenfeld, M. J.: Substantial Seasonal Contribution of Observed Biogenic Sulfate Particles to Cloud Condensation Nuclei, Sci Rep-Uk,
8, 3235, 2018.

Seinfeld, J. H. and Pandis, S. N.: Atmospheric chemistry and physics: from air pollution to climate change, John Wiley & Sons, 2016.

Zheng, G., Wang, Y., Aiken, A. C., Gallo, F., Jensen, M. P., Kollias, P., Kuang, C., Luke, E., Springston, S., Uin, J., Wood, R., and Wang, J.: Marine boundary layer aerosol in the eastern North Atlantic: seasonal variations and key controlling processes, Atmos. Chem. Phys., 18, 17615-17635, 2018.